# Work-Related Flow in Contrast to Either Happiness or PERMA Factors for Human Resources Management Development of Career Sustainability

Carol Nash 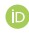

History of Medicine Program, Department of Psychiatry, Temerty Faculty of Medicine, University of Toronto, Toronto, ON M5S 1A1, Canada; carol.nash@utoronto.ca

**Abstract:** In promoting career sustainability, psychological theories historically have informed human resource management (HRM) development—three assessment directions are among them: work-related flow, happiness promotion, and appraising PERMA (Positive Emotions, Engagement, Relationships, Meaning, and Accomplishment) factors. Csikszentmihalyi's work-related flow represents an optimally challenging work-related process. Happiness promotion strives to maintain a pleased satisfaction with the current experience. PERMA represents measurable positive psychological factors constituting well-being. Reliable and validated, the experience of flow has been found to determine career sustainability in contrast to the more often investigated happiness ascertainment or identifying PERMA factors. Career sustainability research to inform HRM development is in its infancy. Therefore, publishers' commitment to sustainability provides integrity. Given MDPI's uniquely founding sustainability concern, its journal articles were searched with the keywords "flow, Csikszentmihalyi, work", excluding those pertaining to education, health, leisure, marketing, non-workers, and spirituality, to determine the utilization of work-related flow to achieve career sustainability. Of the 628 returns, 28 reports were included for potential assessment. Current studies on Csikszentmihalyi's work-related flow ultimately represented three results. These provide insight into successful, positive methods to develop career sustainability. Consequently, HRM is advised to investigate practices for assessing and encouraging employees' engagement with work-related flow with the aim of ensuring career sustainability.

**Keywords:** career sustainability; HRM; work-related flow; happiness; PERMA; Csikszentmihalyi; positive psychology; MDPI

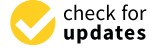



## 1. Introduction

Psychological theories historically have informed the methodology of human resource management (HRM) development regarding career sustainability [1] with psychological assessment being key [2]. Three directions of psychological assessment are notable: work-related flow, happiness promotion, and assessing PERMA (Positive Emotions, Engagement, Relationships, Meaning, and Accomplishment) factors. Csikszentmihalyi's work-related flow is a process of being optimally challenged [3]. Happiness promotion strives to maintain a state of pleased satisfaction with current experience, yet is sometimes associated with system justification, resistance to change, and insensitivity to inequality and social injustice [4]. PERMA represents five measurable factors of positive psychology that may be difficult to differentiate between an overall sense of well-being and day-to-day experiences of living a high-quality life [5]. Reliable and validated, the work-related process of flow has been found to determine career sustainability [6] compared with happiness ascertainment or identifying PERMA factors, neither of which relate to the process and are instead time-specific [7,8]. In this regard, a preference to reduce macrolevel HRM outcomes explanation to individual-level psychological behavioral factors and individual differences

has been recognized as a persistent failing in HRM development [9]. In aiming at career sustainability supportive of satisfying work-related experiences across the working lifespan [10], HRM has been considered to lag in this aim [11] by its focus on assessing factors and differences. Therefore, it is surprising that little research has been conducted into HRM development practices regarding work-related flow, as it is a valuable process enhancing the psychological health that leads to career sustainability [12], in contrast to research on time-dependent happiness or PERMA factors. What research on Csikszentmihalyi's work-related flow is available that might inform HRM development practices to support career sustainability is the focus of undertaking this limited review.

Work-related flow is a theory pioneered by psychologist Mihaly Csikszentmihalyi (1934–2021 [13]) of (1) clear goals every step of the way; (2) immediate feedback to one's actions; (3) a balance between challenges and skills; (4) action and awareness being merged; (5) distractions being excluded from consciousness; (6) no worry of failure; (7) the disappearance of self-consciousness; (8) a distorted sense of time; and (9) the activity becoming autotelic [14]. As a goal-directed process guided by meeting personally developed challenges, flow is identified as the optimal work-related experience [15]. Yet, flow is not equivalent to the experiences of fun, joy, or happiness [16]. This is because aspects of the flow process may be found frustrating by the employee during a particularly challenging task during the process of pursuing the goal [17]. Recognized as one of the most significant in contemporary psychology [18], the importance of this theory—particularly with the 1990 publication of Csikszentmihalyi's book *Flow: The Psychology of Optimal Experience* [19]—was lauded by the then president of the American Psychological Association (1998–2000), Martin Seligman, in describing Csikszentmihalyi as "the world leader in positive psychology research" [18]. With his book translated into twenty-three languages by his death in 2021, *Flow*'s author is among the most cited psychologists in a variety of fields [18], with the theory of flow validated [20] and, apart from refinement [21], remaining virtually unaltered since its development [22].

Happiness in comparison is a judgment-dependent state in which a person, upon considering their situation, determines that they have pleased-satisfaction with their current experience [23] and would like it to continue [24]. Unlike happiness, flow is not a judgment. Instead, it is an engaged process in which the person in flow feels indistinct from their ongoing activity [25]. If a person is asked to evaluate their situation while experiencing flow, rather than describing themselves as happy, the likely response would be that they feel optimally challenged [26], although once the flow experience is over, they could be expected to say that they found the process in its totality enjoyable [27]. It is because happiness is time-limited [28] that development programs by HRM focused on happiness in principle cannot produce career sustainability. In this respect, career sustainability is dependent on developing and maintaining a process, like flow, rather than encouraging the judgment of happiness, which represents one particular point in time.

In 1998, flow theory became the center of psychology that concentrated on "the good life—what it is to be healthy and sane, and what humans choose to pursue when they are not suffering or oppressed" [29] (p. 3), initiated by Seligman and Csikszentmihalyi [29]. From its initial focus on flow, positive psychology was then redirected in 2011 based on Seligman's suggestion that PERMA (Positive Emotion, Engagement, Relationships, Meaning, and Accomplishment) factors constitute well-being [30]. Although the originality of this measure of the good life has been called into question as differing little from the 1984 three-factor model of subjective well-being [31], PERMA factors have become a contemporary priority of research in positive psychology [32–34], including HRM research [35]. Yet, studies supporting this model have been called into question by a systematic review of PERMA related to HRM [36] and by a thirty-year positive psychology researcher's findings that these measures remain poorly constructed [37]. For these reasons, there are conceptual difficulties with basing HRM development on PERMA factors for career sustainability. Contrasted to these PERMA factors, flow has continuously demonstrated validity and reliability [38–40] with respect to career sustainability.

Career sustainability is a form of human sustainability in its early stages of development [41]. As such, research regarding career sustainability gains important foundational support through publications acknowledging devotion to sustainability. Comparing how flow has been referenced in contrast to happiness promotion or PERMA factors in articles published in journals publicly committed to sustainability is thus the focus of this investigation. MDPI (Multidisciplinary Digital Publishing Institute) was founded as a publisher of open access scientific journals and a member of the United Nations Global Compact in support of corporate sustainability [42]. Sustainability is stated as at the core of MDPI's values [43], with a flagship journal devoted to sustainability as well as organized conferences and events based on sustainability [44]. Although other publishing houses have adopted this commitment [45–47], MDPI appears to be the only publishing house founded with sustainability as its focus. As this study concerns creating a foundation for research on career sustainability, the importance of sustainability for the publishing house regarding its publications was deemed valuable for ensuring research integrity.

Predicated on MDPI's uniquely founding sustainability concern as an important indicator of concern for career sustainability, a search was conducted using the MDPI search engine on 22 December 2023 for all MDPI articles with the keywords "flow, Csikszentmihalyi, work". The purpose of the search was to determine how Csikszentmihalyi's work on flow has been interpreted by MDPI authors regarding career sustainability in light of the publisher's commitment to sustainability in comparison with mentions made of happiness and PERMA in these same articles.

The role of HRM development is predicated on motivating employees to stay energized in order for firms to survive and thrive through adverse conditions, with HRM seeking to incorporate employees' positive psychological resources to successfully contend with uncertainties and crises [48]. Although interest in flow remains regarding career sustainability, research focus in the MDPI publications returned in the search conducted has noted a preference for investigations regarding happiness and PERMA factors even when flow is referenced. Nevertheless, based on the research results returned regarding flow, HRM is presented with practical methods to support the development of work-related flow that have been demonstrated to produce career sustainability in contrast to a focus on either happiness promotion or measuring PERMA factors at a point in time.

## 2. Materials and Methods

The materials to be gathered will follow a limited review. As such, this is neither a systematic review nor a scoping review—technical terms concerning the Preferred Reporting Items for Systematic Reviews and Meta-Analyses (PRISMA) [49]. Systematic reviews investigate predefined research questions using clear, reproducible methods to identify, critically appraise, and assess the results of primary research studies. Key stages in the systematic review process include the clarification of aims and methods in a protocol, searching for relevant research, collecting data, assessing the quality of the studies returned, synthesizing evidence, and interpreting findings quantitatively and qualitatively [50]. Although a similar process was followed to gather the materials for this study as used in a systematic review, the search undertaken was not equivalent to a systematic review as the protocol was not registered, and no quantitative statistical analysis was performed. Furthermore, this limited review is not a scoping review because only MDPI publications were searched rather than a full range of databases, and the keywords searched were not extensive enough to make the search comprehensive [50]. Nevertheless, the PRISMA process was selected for use in this limited review as it has been identified as involving high-level procedures [51]. In making use of the PRISMA diagram for this limited review, it should not be interpreted that this review is claiming to be either a systematic or scoping review for the above-mentioned reasons. Why a limited review was chosen has been stated in the Introduction and concerns the unique founding commitment of MDPI as a publishing house devoted to sustainability.

A 22 December 2023 search of MDPI articles with the keywords "flow, Csikszentmihalyi, work" was conducted. The results returned represent the materials of this study, while

the process undertaken, depicted in the flow diagram of Figure 1, demonstrates the methods. The records identified numbered 628, with 2 removed for lacking Csikszentmihalyi. The following categories are offered as those excluded for not pertaining to work—plus their number: education—171; health—96; leisure—101; marketing—56; non-worker—123; and spirituality—51. The reports of the included studies following these exclusions total 28, representing the materials for which results will be provided.

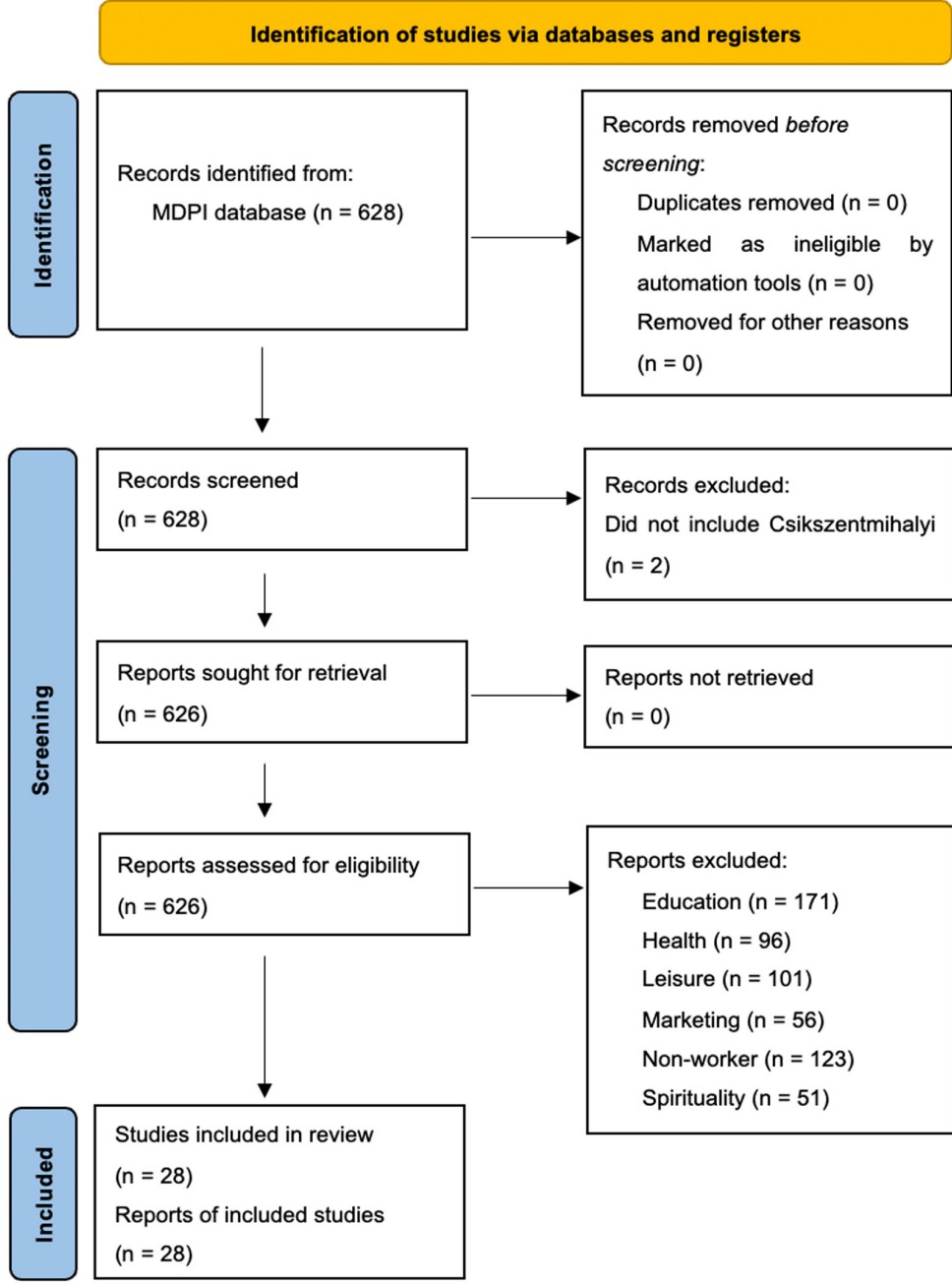

**Figure 1.** Results of a 22 December 2023 MDPI search of keywords "flow, Csikszentmihalyi, work". The flowchart template follows PRISMA recommendations [49] for systematic and scoping reviews.

## 3. Results

The reports included in the search performed are the articles mentioning work-related flow while referencing at least one publication by Csikszentmihalyi. These cited articles are examined in the following manner: first, year of publication and the MDPI journal title (see Table 1). Considering that MDPI began publishing in 1996 [43], there was no restriction

placed on the year of publication for the search conducted. As such, reports that were included might have returned those older than five years, yet the best practice in scientific research recommends that references be published within the last five years [52]. Therefore, although all reports of included studies will be listed, only those reports published within the last five years will be discussed—those from 2019–2023. Second, the articles cited, written by Csikszentmihalyi, will be examined for how flow is referenced by the authors of the reports included. The data for this second examination are found in Table 2. Third to be considered is whether the understanding of flow in the article is focused on flow, happiness, or some other measure, with particular attention to whether PERMA factors are the "other" mentioned. The data for this assessment are found in Table 3.

**Table 1.** Included reports for a 22 December 2023 search of MDPI articles with the keywords "flow, Csikszentmihalyi, work" regarding their citation number, the truncated article title, the year of publication, and the MDPI journal in which the article is published.

| # | Article Title (Truncated) | Year | MDPI Journal Title |
|---|---|---|---|
| 53 | Building Work Engagement in Organizations | 2023 | Behavioral Sciences |
| 54 | Changing the Environment Based on Empowerment | 2014 | Entropy |
| 55 | Contributions to Sustainability in SMEs: Human Resources | 2021 | Sustainability |
| 56 | Creativity and Resilience as Predictors of Career Success | 2021 | Sustainability |
| 57 | Does Happiness Launch More Businesses? Affect, Gender | 2020 | International Journal of Environmental Research and Public Health |
| 58 | Effect of Attainment Value and Positive Thinking | 2020 | Journal of Open Innovation: Technology, Market, and Complexity |
| 59 | Effects of Selected Positive Resources on Hospitality Service | 2019 | Sustainability |
| 60 | Engagement, Passion, and Meaning of Work | 2019 | International Journal of Environmental Research and Public Health |
| 61 | Explaining the Paradox: How Pro-Environmental | 2013 | Sustainability |
| 62 | Factors Affecting Entrepreneurship and Business | 2018 | Sustainability |
| 63 | Gratitude in Organizations: Psychometric Properties | 2022 | International Journal of Environmental Research and Public Health |
| 64 | Human Resource Practices, Eudaimonic Well-Being | 2019 | Sustainability |
| 65 | Intelligence and Creativity: Mapping Constructs | 2020 | Journal of Intelligence |
| 66 | Occupational Stress: Preventing Suffering, Enhancing Wellbeing | 2016 | International Journal of Environmental Research and Public Health |
| 67 | Personality Traits and Positive Resources of Workers | 2018 | Sustainability |
| 68 | Positive Orientation and Strategies for Coping with Stress | 2019 | International Journal of Environmental Research and Public Health |
| 69 | Promoting Flow at Work through Proactive Personality | 2022 | Sustainability |
| 70 | Psychological Capital, Workload, and Burnout | 2020 | Sustainability |
| 71 | Relationships between High Ability (Gifted) and Flow | 2020 | Sustainability |
| 72 | Revisiting the Happy-Productive Worker Thesis | 2021 | Sustainability |
| 73 | David J. Rowe's Career-Long Methods | 2022 | Challenges |
| 74 | Self-Perceived Employability and Meaningful Work | 2019 | Sustainability |
| 75 | 1The Influence of Corporate Social Responsibility | 2018 | Sustainability |
| 76 | The Longitudinal Link Between Organizational Citizenship | 2021 | International Journal of Environmental Research and Public Health |
| 77 | The Nature of Job Crafting: Positive and Negative Relations | 2019 | International Journal of Environmental Research and Public Health |
| 78 | The Role of Relationships at Work and Happiness | 2019 | Sustainability |
| 79 | To Be Happy: A Case Study of Entrepreneurial | 2020 | Sustainability |
| 80 | What Are the Common Themes of Physician Resilience? | 2022 | International Journal of Environmental Research and Public Health |

**Table 2.** Included reports for a 22 December 2023 search of MDPI articles with the keywords "flow, Csikszentmihalyi, work" regarding their citation number in this publication, the truncated article title, the citation number in this publication of the work by Csikszentmihalyi cited in the included report, and the truncated title of the work by Csikszentmihalyi cited by the included report.

| # | Article Title (Truncated) | # | Csikszentmihalyi Title (Truncated) |
|---|---|---|---|
| 53 | Building Work Engagement in Organizations | 81 | Positive Psychology. An Introduction |
| 54 | Changing the Environment Based on Empowerment | 27 | Beyond Boredom and Anxiety |
| 55 | Contributions to Sustainability in SMEs: Human Resources | 82 | Creatividad: El Fluir Y La Psicología |
| 56 | Creativity and Resilience as Predictors of Career Success | 82 | Creatividad: El Fluir y La Psicología |
| 57 | Does Happiness Launch More Businesses? Affect, Gender | 19 | Flow |
| 58 | Effect of Attainment Value and Positive Thinking | 83 | Optimal Experience: Psychological |
| 59 | Effects of Selected Positive Resources on Hospitality Service | 81 | Positive psychology: An introduction |
| 60 | Engagement, Passion, and Meaning of Work | 81 | Positive psychology: An introduction |
| 61 | Explaining the Paradox: How Pro-Environmental | 84 | If we are so rich, why aren't we happy? |
| 62 | Factors Affecting Entrepreneurship and Business | 85 | The concept of flow |
| 63 | Gratitude in Organizations: Psychometric Properties | 81 | Positive psychology: An introduction |
| 64 | Human Resource Practices, Eudaimonic Well-Being | 81 | Positive psychology: An introduction |
| 65 | Intelligence and Creativity: Mapping Constructs | 86 | The experience sampling method |
| | | 87 | All You Need Is Love |
| 66 | Occupational Stress: Preventing Suffering, Enhancing Wellbeing | 81 | Positive psychology: An introduction |
| 67 | Personality Traits and Positive Resources of Workers | 81 | Positive psychology: An introduction |
| 68 | Positive Orientation and Strategies for Coping with Stress | 81 | Positive psychology: An introduction |
| 69 | Promoting Flow at Work through Proactive Personality | 27 | Beyond boredom and anxiety |
| 70 | Psychological Capital, Workload, and Burnout | 81 | Positive psychology: An introduction |
| 71 | Relationships between High Ability (Gifted) and Flow | 88 | The Creative Vision |
| | | 82 | Creatividad. El Fluir y la Psicología |
| | | 19 | Flow |
| | | 89 | Optimal experience in work and leisure |
| | | 14 | Creativity: The Work and Lives * |
| | | 90 | Musical improvisation |
| | | 91 | Proneness for Psychological Flow |
| | | 92 | Cultivating talent throughout life |
| | | 93 | Flow theory and research |
| 72 | Revisiting the Happy-Productive Worker Thesis | 19 | Flow |
| 73 | David J. Rowe's Career-Long Methods | 19 | Flow |
| | | 84 | If we are so rich, why aren't we happy? |
| | | 89 | Optimal experience in work and leisure |
| | | 18 | Optimal experience in adult learning |
| 74 | Self-Perceived Employability and Meaningful Work | 94 | The construction of meaning |
| 75 | The Influence of Corporate Social Responsibility | 81 | Positive psychology: An introduction |
| 76 | The Longitudinal Link Between Organizational Citizenship | 81 | Positive psychology: An introduction |
| 77 | The Nature of Job Crafting: Positive and Negative Relations | 19 | Flow |
| 78 | The Role of Relationships at Work and Happiness | 81 | Positive psychology: An introduction |
| 79 | To Be Happy: A Case Study of Entrepreneurial | 81 | Positive psychology: An introduction |
| 80 | What Are the Common Themes of Physician Resilience? | 19 | Flow |

*\* Creativity: The Work and Lives of 91 Eminent People* is now retitled *Creativity: Flow and the Psychology of Discovery and Invention*. It is the same book with the same ISBN.

**Table 3.** Reports of included studies for a 22 December 2023 search of MDPI articles with the keywords "flow, Csikszentmihalyi, work" regarding their citation number in this document, the truncated article title, and whether the article mentions, flow, happiness or some other aspect to develop career sustainability noting those that refer to one or more PERMA factors. "✗" denotes no mention; " ✓ " indicates mention. Items emboldened represent the reports meeting the criteria for assessment.

| # | Article Title (Truncated) | Flow | Happiness | Other |
|---|---|---|---|---|
| 53 | Building Work Engagement in Organizations | ✗ | ✗ | Work engagement * |
| 54 | Changing the Environment Based on Empowerment | ✓ | ✗ | ✗ |

**Table 3.** *Cont.*

| # | Article Title (Truncated) | Flow | Happiness | Other |
|---|---|---|---|---|
| 55 | **Contributions to Sustainability in SMEs: Human Resources** | ✗ | ✗ | **Creativity** |
| 56 | **Creativity and Resilience as Predictors of Career Success** | ✗ | ✓ | **Creativity** |
| 57 | **Does Happiness Launch More Businesses? Affect, Gender** | ✗ | ✓ | **Positive emotions \*** |
| 58 | **Effect of Attainment Value and Positive Thinking** | ✗ | ✓ | **Engagement \*** <br> **Positive emotions \*** <br> **Creativity** |
| 59 | Effects of Selected Positive Resources on Hospitality Service | ✗ | ✗ | Work engagement \* <br> Positive emotions \* |
| 60 | Engagement, Passion, and Meaning of Work | ✓ | ✓ | Positive emotions \* <br> Character strengths |
| 61 | Explaining the Paradox: How Pro-Environmental | ✗ | ✓ | Doing good |
| 62 | Factors Affecting Entrepreneurship and Business | ✓ | ✓ | Motivation |
| 63 | Gratitude in Organizations: Psychometric Properties | ✗ | ✓ | Gratitude |
| 64 | Human Resource Practices, Eudaimonic Well-Being | ✗ | ✓ | Creativity |
| 65 | Intelligence and Creativity: Mapping Constructs | ✗ | ✓ | Creativity |
| 66 | Occupational Stress: Preventing Suffering, Enhancing Wellbeing | ✗ | ✓ | Engagement \* <br> Positive emotions \* <br> Satisfaction <br> Meaning \* |
| 67 | Personality Traits and Positive Resources of Workers | ✗ | ✗ | Optimism <br> Hope |
| 68 | Positive Orientation and Strategies for Coping with Stress | ✗ | ✗ | Positive orientation |
| 69 | **Promoting Flow at Work through Proactive Personality** | ✓ | ✓ | **Engagement \*** |
| 70 | Psychological Capital, Workload, and Burnout: | ✗ | ✗ | Psychological capital |
| 71 | **Relationships between High Ability (Gifted) and Flow** | ✓ | ✗ | **✗** |
| 72 | **Revisiting the Happy-Productive Worker Thesis** | ✗ | ✗ | **Engagement \*** |
| 73 | **David J. Rowe's Career-Long Methods** | ✓ | ✓ | **✗** |
| 74 | **Self-Perceived Employability and Meaningful Work** | ✗ | ✓ | **Meaningful work \*** |
| 75 | The Influence of Corporate Social Responsibility | ✗ | ✓ | Meaningful work \* |
| 76 | The Longitudinal Link Between Organizational Citizenship | ✗ | ✓ | Flourishing † |
| 77 | **The Nature of Job Crafting: Positive and Negative Relations** | ✗ | ✗ | **Job autonomy** |
| 78 | The Role of Relationships at Work and Happiness | ✗ | ✓ | Meaningful work \* |
| 79 | To Be Happy: A Case Study of Entrepreneurial | ✗ | ✓ | Entrepreneurship |
| 80 | **What Are the Common Themes of Physician Resilience?** | ✗ | ✓ | **Self-determination** |

\* A PERMA factor (Positive Emotions, Engagement, Relationships, Meaning, or Accomplishment). † Article directly references PERMA.

The truncated titles of the reports included represent 28 MDPI articles. They are as follows: Building Work Engagement in Organizations [53], Changing the Environment Based on Empowerment [54], Contributions to Sustainability in SMEs: Human Resources [55], Creativity and Resilience as Predictors of Career Success [56], Does Happiness Launch More Businesses? Affect, Gender [57], Effect of Attainment Value and Positive Thinking [58], Effects of Selected Positive Resources on Hospitality Service [59], Engagement, Passion and Meaning of Work [60], Explaining the Paradox: How Pro-Environmental [61], Factors Affecting Entrepreneurship and Business [62], Gratitude in Organizations: Psychometric Properties [63], Human Resource Practices, Eudaimonic Well-Being [64], Intelligence and Creativity: Mapping Constructs [65], Occupational Stress: Preventing Suffering, Enhancing Wellbeing [66], Personality Traits and Positive Resources of Workers [67], Positive Orientation and Strategies for Coping with Stress [68], Promoting Flow at Work through Proactive Personality [69], Psychological Capital, Workload, and Burnout [70], Relationships between High Ability (Gifted) and Flow [71], Revisiting the Happy-Productive Worker Thesis [72], David J. Rowe's Career-Long Methods [73], Self-Perceived Employability and Meaningful Work [74], The Influence of Corporate Social Responsibility [75], The Longitudinal Link between Organizational Citizenship [76], The Nature of Job Crafting: Positive and Negative Relations [77], The Role of Relationships at Work and Happiness [78], To Be Happy: A Case Study of Entrepreneurial [79], What Are the Common Themes of Physician Resilience? [80].

### 3.1. Publication Date and Journal Title

The publication dates of the reports included for assessment citing Csikszentmihalyi concerning work-related flow activities, as well as the journal titles of those publications, are listed in Table 1.

As *Flow* was published in 1990 [19], and the MDPI publishing house began in 1996 [43], that the first article published regarding the search for "flow, Csikszentmihalyi, work" was in 2013 is noteworthy. Although the theory of work-related flow was available for research assessment, 17 years passed before a paper was published in an MDPI journal citing Csikszentmihalyi in this regard. Furthermore, the majority of the returned articles were published in 2019 and 2020. Older than five years, the following six articles will not be considered in the Discussion to follow [54,61,62,66,67,75]. Figure 2 illustrates the publication year and number of publications per year.

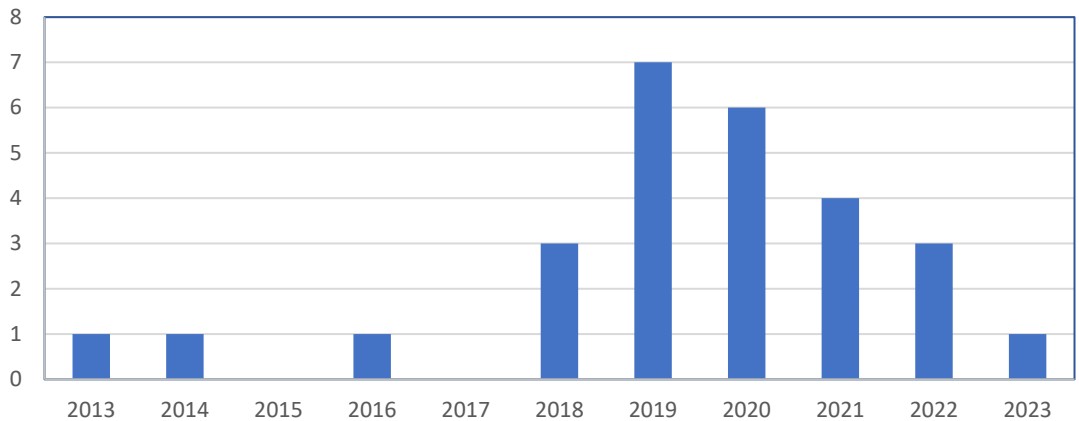

**Figure 2.** Number of publications in each year of publication for all articles in MDPI journals referencing Csikszentmihalyi concerning work-related flow.

Figure 3 depicts the number of publications that reference Csikszentmihalyi in each of the various journals in which articles on work-related flow were published during the history of the MDPI publishing house. It is relevant that the largest number of publications have been published in MDPI's premiere journal specifically regarding sustainability [44], as this demonstrates that these various publications represent the aim of the publishing house as intended [43] most directly.

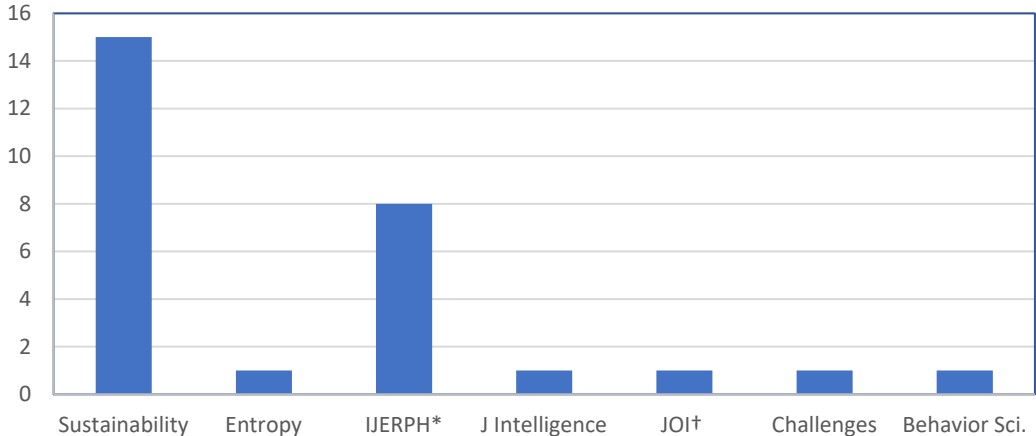

**Figure 3.** MDPI journal names publishing articles referencing Csikszentmihalyi regarding work-related flow depicted in the order of the date of the first publication in the various journals. * International Journal of Environmental Research and Public Health, † Journal of Open Innovation: Technology, Market, and Complexity.

### 3.2. Reference to Csikszentmihalyi Articles

Table 2 lists the Csikszentmihalyi articles referenced by the reports returned in the 22 December 2023 keyword search of "flow, Csikszentmihalyi, work". Some articles contain more than one reference to a Csikszentmihalyi publication, so each is listed. There are 18 different Csikszentmihalyi references that historically MDPI journal articles have cited. These include the following titles: Positive Psychology: An Introduction [81], *Beyond Boredom and Anxiety: Experiencing Flow in Work and Play* [27], *Creatividad: El Fluir y la Psicología del Descubirmiento y la Invención* [82], *Flow: The Psychology of Optimal Experience* [19], *Optimal Experience: Psychological Studies of Flow in Consciousness* [83]. If We Are So Rich, Why Aren't We Happy? [84], The Concept of Flow [85], The Experience Sampling Method [86], All You Need Is Love: The Importance of Partner and Family Relations to Highly Creative Individuals' Well-Being and Success [87], *The Creative Vision: a Longitudinal Study of Problem Finding in Art* [88], Optimal Experience in Work and Leisure [89], *Creativity: Flow and the Psychology of Discovery and Invention* [14], Musical Improvisation: A Systems Approach [90], Proneness for Psychological Flow [91], Cultivating Talent Throughout Life [92], Flow Theory and Research [93], Optimal Experience in Adult Learning: Conception and Validation of the Flow in Education Scale (EduFlow-2) [18], The Construction of Meaning Through Vital Engagement [94].

The particular concern of this investigation is those articles that not only reference Csikszentmihalyi but that they do so regarding his theory of flow in a work-related setting. Figure 4 demonstrates that 13 of the 28 articles that cite Csikszentmihalyi are in regard to his article, Positive Psychology: An Introduction. Although the theory of flow is mentioned in this publication, there is no explanation of the theory concerning work. As such, in the Discussion to follow, those of the 28 articles that cite only this article regarding work-related flow will be excluded from consideration. Beyond those that have been eliminated already from the Discussion to follow as a result of being published pre-2019 [54,61,62,66,67,75] these additional 11 records will not be part of the Discussion because they reference Positive Psychology: An Introduction and no other work by Csikszentmihalyi [53,59,60,63,66–68,75,76,78,79].

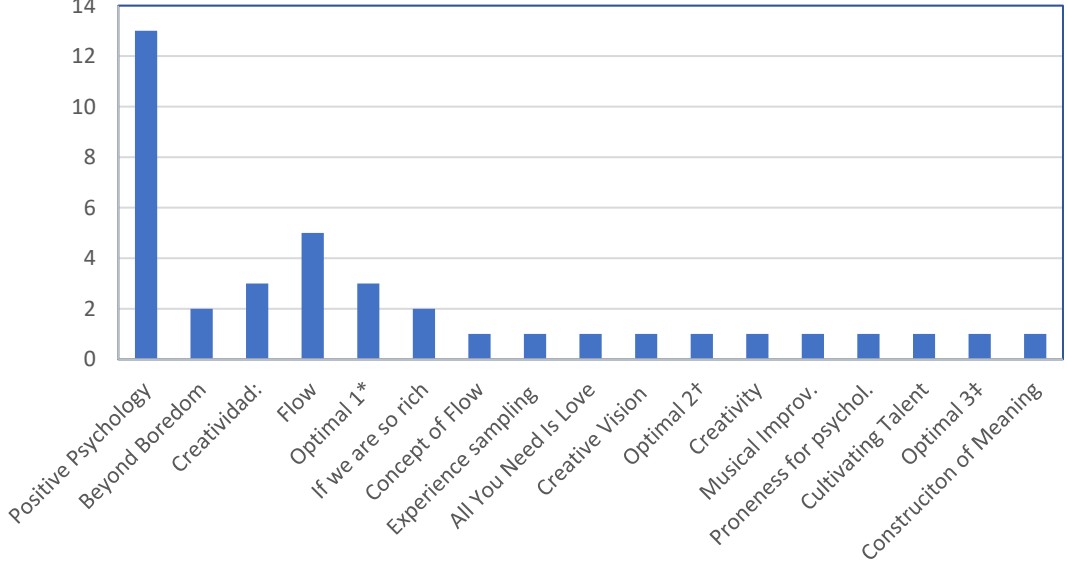

**Figure 4.** Number and title of Csikszentmihalyi publications referenced in the various MDPI articles in order of their appearance in Table 2. * Optimal Experience: Psychological Studies of Flow in Consciousness, † Optimal Experience in Work and Leisure, ‡ Optimal Experience in Adult Learning: Conception and Validation of the Flow in Education Scale (EduFlow-2).

Further to be eliminated from the assessment are those returned records that reference other Csikszentmihalyi-authored publications that do not discuss the theory of work-

related flow. This includes one record [64]. It references publications by Csikszentmihalyi related to daily living practices rather than to work [86,87].

*3.3. Flow, Happiness, or Other (Including PERMA Factors)*

Table 3 represents whether the included reports focus on flow, happiness, or another measure (with specific reference to PERMA factors if they are the other). Although the search undertaken was regarding the keywords "flow, Csikszentmihalyi, work" an important result to note is that a greater number of these reports returned focused on either happiness or at least one of the PERMA factors rather than flow. This result is an indication of the importance of referencing happiness, and PERMA factors even in works that reference Csikszentmihalyi 's theory of flow. In total, 18 of the 28 articles were directed towards happiness, while 12 of the 28 focused on one PERMA factor or more. There were 9 articles that referred to both happiness and at least one PERMA factor, and 2 that discussed each flow, happiness, and a PERMA factor [60,69]. In the case of the first article of the two, the PERMA factor mentioned is Positive emotions. The second article, concerns the PERMA factor Engagement. What is also interesting is that there were only 2 articles returned that concentrated on an examination of flow alone [54,71], one of which will not be considered in the Discussion because it was published in 2014 [54].

The articles that will no longer be considered have been eliminated because they are either too old [54,61,62,66,67,75] or reference works by Csikszentmihalyi that do not focus on the theory of work-related flow [53,59,60,63,66–68,75,76,78,79]. With these articles now eliminated from further consideration, only those articles emboldened in Table 3 will be those referred to in the Discussion to follow [55–58,69,71–74,77,80].

*3.4. Final Reports Included to Be Considered for Assessment*

As listed in Table 4, the final reports to be considered for assessment of those included are 11 in number [55–58,69,71–74,77,80]. Of these, 6 articles are published in *Sustainability*, 3 in the *International Journal of Environmental Research and Public Health*, and 1 each in the *Journal of Open Innovation: Technology, Market, and Complexity*, and in *Challenges*. The number of Csikszentmihalyi references cited by these reports in total equals 13. They are cited here in relation to how they are cited in Table 4 [14,18,19,82–84,88–94]. Of these 11 reports, 3 mention flow, 7 consider happiness, and 9 investigate some other measure, including identifying PERMA factors in 6 cases (one report mentioned two of the PERMA factors).

**Table 4.** Final 11 reports to be considered for assessment of included studies for a 22 December 2023 search of MDPI articles with the keywords "flow, Csikszentmihalyi, work" regarding their citation number, the MDPI journal in which they are published, the work by Csikszentmihalyi that is referenced, and whether the article mentions, flow, happiness or some other aspect to develop career sustainability noting those that refer to one or more PERMA factors. "✗" denotes no mention; " ✓ " indicates mention.

| # | MDPI Journal Title | # | Csikszentmihalyi Title (Truncated) | Flow | Happiness | Other |
|---|---|---|---|---|---|---|
| 55 | Sustainability | 82 | Creatividad: El Fluir Y La Psicología | ✗ | ✗ | Creativity |
| 56 | Sustainability | 82 | Creatividad: El Fluir Y La Psicología | ✗ | ✓ | Creativity |
| 57 | International Journal of Environmental Research and Public Health | 19 | Flow | ✗ | ✓ | Positive emotions * |
| 58 | Journal of Open Innovation: Technology, Market, and Complexity | 83 | Optimal Experience: Psychological | ✗ | ✓ | Engagement * Positive emotions * Creativity |
| 69 | Sustainability | 27 | Beyond boredom and anxiety | ✓ | ✓ | Engagement * |

**Table 4.** *Cont.*

| # | MDPI Journal Title | # | Csikszentmihalyi Title (Truncated) | Flow | Happiness | Other |
|---|---|---|---|---|---|---|
| 71 | Sustainability | 88 | The Creative Vision | ✓ | ✗ | ✗ |
| | | 82 | Creatividad. El Fluir y la Psicología | | | |
| | | 19 | Flow | | | |
| | | 89 | Optimal experience in work and leisure | | | |
| | | 14 | Creativity: The Work and Lives | | | |
| | | 90 | Musical improvisation | | | |
| | | 91 | Proneness for Psychological Flow | | | |
| | | 92 | Cultivating talent throughout life | | | |
| | | 93 | Flow theory and research | | | |
| 72 | Sustainability | 19 | Flow | ✗ | ✗ | Engagement * |
| 73 | Challenges | 19 | Flow | ✓ | ✓ | ✗ |
| | | 84 | If we are so rich, why aren't we happy? | | | |
| | | 89 | Optimal experience in work and leisure | | | |
| | | 18 | Optimal experience in adult learning | | | |
| 74 | Sustainability | 94 | The construction of meaning | ✗ | ✓ | Meaningful work * |
| 77 | International Journal of Environmental Research and Public Health | 19 | Flow | ✗ | ✗ | Job autonomy |
| 80 | International Journal of Environmental Research and Public Health | 19 | Flow | ✗ | ✓ | Self-determination |

* A PERMA factor (Positive Emotions, Engagement, Relationships, Meaning, or Accomplishment).

## 4. Discussion

This work aims to determine how work-related flow has been referenced in contrast to happiness or PERMA factors in articles published since 2019 in journals publicly committed to sustainability whereas MDPI journals have been noted as those specifically created in relation to sustainability. Why flow is the focus is that PERMA factor measurement has been found unreliable regarding career sustainability [36,37] and a focus on happiness is concerning a fixed time point rather than consistent work sustainability [23,24]. Although flow has been referenced in each of the 11 final reports considered for assessment, only three of the articles were found to actually discuss flow as important to career sustainability [69,71,73]. As a result, only these three reports are further assessed. Two of these articles discuss both flow and happiness [69,73], while one concerns flow alone [71]. Of these reports, two are published in *Sustainability* [69,71], while one is published in *Challenges* [73]. One of these included reports also discusses a PERMA factor [69]; this is an article published in *Sustainability*. The relevance of the points made in these three papers concerning HRM of career sustainability is discussed next.

### 4.1. Three Reports Discussing Flow

The three reports published within the last five years regarding work-related flow are examined regarding when and where they were published, the work by Csikszentmihalyi they reference, the aim of the study, its results, how it interprets work-related flow, and any mention is made of either happiness or a PERMA factor.

#### 4.1.1. Proactive Personality and Flow in Italian Employees

The full title of the first article regarding flow is Promoting Flow at Work through Proactive Personality: A Sequential Mediation Model with Evidence from Italian Employees [69]. It was published in 2022 in *Sustainability*. The Csikszentmihalyi work that the authors rely on regarding their understanding of work-related flow is his book published in 2000, *Beyond Boredom and Anxiety: Experiencing Flow in Work and Play* [27]. The authors seek to fill a gap in the literature regarding the relationship between proactivity and flow at work, and the role of the proactive personality, job crafting, and work engagement (a PERMA factor) in promoting flow at work. They report on an online study of 362 Italian employees to determine the relationship among these variables. The results indicate that

proactive employees are able to experience flow at work through job crafting and work engagement. However, the value of these results related to Csikszentmihalyi's meaning of flow can be questioned. The reason is that after introducing Csikszentmihalyi's definition of flow [27], the authors turn to an alternative definition of the term [95] depending on the work of Ryan and Deci regarding self-determination theory [96]—which deals with evaluations of individual action (e.g., acting autonomously or competently)—in contrast to flow theory which focuses on the experience itself [97]. In substituting another understanding of flow than Csikszentmihalyi's, these authors have interpreted the purpose of engaging in flow to be enjoyment. As has been noted, the focus of flow in Csikszentmihalyi's research findings is not enjoyment (although he believed life's activities should be enjoyable [98] (pp. xx–xxi)), rather employees engage in flow and prefer flow activities because they are optimally challenging and this is what produces the optimal work-related experience [89]. Consequently, it is unclear whether those with a proactive personality are most likely to craft work that produces the type of work engagement that is optimally challenging. Rather, their focus may be making their work as enjoyable as possible. These authors have reinterpreted flow. This matters because they consider flow a short-term peak experience. Interpreted in this way, contrary to Csikszentmihalyi's understanding, flow is not career-sustainable. For this reason, the results of this study are not obviously relevant regarding HRM developing career sustainability. How the authors refer to happiness and PERMA factors is regarding their relationship stating, "employees with high levels of engagement tend to feel positive emotions at work (for instance, happiness, joy, and enthusiasm), subsequently achieving good performance". There is no further discussion of either happiness or work engagement in the article.

### 4.1.2. Flow in High-Ability Spanish Music Performers

The second article that concerns flow is titled Relationships between High Ability (Gifted) and Flow in Music Performers: Pilot Study Results [71] and was published in 2020, also in *Sustainability*. The authors of this article reference nine of Csikszentmihalyi's publications: *The Creative Vision: A Longitudinal Study of Problem Finding in Art* [88]; *Creatividad: El Fluir y la Psicología del Descubirmiento y la Invención* [82]; *Flow: The Psychology of Optimal Experience* [83]; Optimal experience in work and leisure [89]; *Creativity: The Work and Lives of 91 Eminent People* (note, the title has now changed for this work) [14]; Musical Improvisation: A Systems Approach [90]; Proneness for Psychological Flow in Everyday Life: Associations with Personality and Intelligence [91]; Cultivating Talent Throughout Life [92]; Flow Theory and Research [93]. Unlike the first article on Italian employees, these authors represent a deeper and more nuanced understanding of Csikszentmihalyi's theory of flow, yet—even with this better understanding—rather than stating that work in this regard is optimally challenging, they instead summarize it as being pleasurable. On the other hand, apart from this summary by the authors, the body of the paper clarifies that their definition of flow is the one supported by Csikszentmihalyi. Their study aims to investigate the relationship between high-ability Spanish musicians, dedication to music, and flow while they engage in musical activities. The result: the best indicator of the flow state is found to a statistically significant degree to be a loss of self-consciousness when performing. What the authors also found is that their results support Csikszentmihalyi's findings that there is no relationship between intelligence and the experience of flow measured using a fluid intelligence test, such as Raven's SPM Plus or the Wiener Matrizen Test [91]. Their results also note that those with the greatest talent experience flow more often than others. In this regard, talent includes the courage to be different, possessing independent thought and action, and having high levels of self-confidence. Additionally, the authors suggest that the flow state during musical performance may relate to greater emotional intelligence based on a 2013 reference [99]. Generally, these authors consider investigating flow to be worthwhile regarding career sustainability, especially concerning the aspect of a loss of self-consciousness, which they consider may be tied to better learning in highly accomplished

musicians. This article discusses work-related flow alone and makes no mention of either happiness or PERMA factors.

### 4.1.3. Promoting Flow in Canadian Physics Research

The final article of the three discussing flow in work is one by this author, published in 2022 in *Challenges.* Its full title is Self-Direction in Physics Graduate Education: Insights for STEM from David J. Rowe's Career-Long Methods [73]. Although this is an essay that concerns education—and search returns regarding education were initially excluded from consideration—rather than education, this publication is focused on career sustainability in the higher education setting. For this reason, it is included in the assessment. The works by Csikszentmihalyi referenced in the paper are: *Flow: The Psychology of Optimal Experience* [19]; If We Are So Rich, Why Aren't We Happy? [84]; Optimal Experience in Work and Leisure [89]; Optimal Experience in Adult Learning: Conception and Validation of the Flow in Education Scale (EduFlow-2) [18]. This study explores the self-directed learning promoted by University of Toronto physicist David J. Rowe in response to the type of research flow he consistently experienced throughout his career as a theoretical mathematical physicist that brought him the greatest work-related satisfaction. Explored are the ways he structured his work with colleagues to encourage everyday flow in their interactions concerning what he provided to them with regard to his space, time, open-mindedness, and the particular theoretical contributions he offered. The results of his methods encouraged his collaborators to sustain their careers through consistently pioneering insightful resolutions to complex, multidimensional, mathematical physics problems. This research process permitted and encouraged Rowe to sustain his own research career until the day of his death at 84. In this regard, Rowe understood work-related flow as Csikszentmihalyi defined it—a desired activity stretching the researcher's mind to its limits in a voluntary effort to accomplish something personally valued as both difficult and worthwhile [83]. In this work, the reference made to happiness is, "as has been reported by those who have experienced flow in their work, it is flow in conducting this research that brings the greatest happiness to researchers". This point is supported by reference to Csikszentmihalyi's publication [84] that, when researchers are asked to assess their flow experience overall at a particular time point, they evaluate it as producing the peak experience of happiness.

### 4.2. Future Research Directions

This study of searched reports of the keywords "flow, Csikszentmihalyi, work" in MDPI journals, investigated for their fundamental support of sustainability, has resulted in the finding that there are only three articles published in MDPI journals within the last five years that investigate how flow might contribute to career sustainability. This paucity of publications on the topic together with the result that work-related flow is recognized as most important for career sustainability because it aligns with employees' values [100] present reasons why the psychological theory of work-related flow as envisioned by Csikszentmihalyi is a useful area in need of research in HRM. In undertaking such work, researchers must be cautious to consider flow as an optimally challenging work-related activity rather than erroneously focusing on it as peak enjoyment. As noted, concerning the Italian study that based its understanding of flow on later research other than that of Csikszentmihalyi, if flow is interpreted as outstanding enjoyment, it is a short-term result rather than a career-sustaining activity. In this way, what needs to be considered by HRM in investigating flow is what can be enhanced in employees and what types of resources can be provided to promote work-related flow.

The research regarding high-ability Spanish musicians recognized high emotional intelligence coupled with a lack of self-consciousness as the areas to be enhanced to encourage flow. In that these researchers presented Csikszentmihalyi's understanding of flow rather than another interpretation, their findings have particular applicability, and the standardized tests for assessing emotional intelligence and a lack of self-consciousness in employees are particularly relevant [101,102].

Regarding how physicist David Rowe organized his work with colleagues to increase their self-direction and the regular research flow of each participant—he followed a six-step process (involving four different aspects) summarized as [73]:

1. Identify resources needed to achieve each goal—(space)
2. Identify the structure and sequence of learning activities—(time)
3. Outline how it will be known those goals have been achieved—(time)
4. Create a timeline for activities' completion—(time)
5. Locate a mentor to provide feedback on the plan—(open-mindedness)
6. Develop goals for study—(theoretical contributions).

That this process developed by Rowe has been successful in promoting consistent work-related flow in his colleagues is witnessed by the comments made by four of the physicists he worked with throughout his career in their adopting similar techniques to those of Rowe in their own research groups. Concerning the influence of Rowe's research program organization, Jerry Draayer, Professor of Physics and Louisiana State University Distinguished Research Master, has stated, "David Rowe was a master mentor and clever innovator—at LSU we are attempting to keep that spirit alive" [103]. George Rosensteel Professor of Physics at Tulane University was Rowe's most illustrious graduate student (who died a few months after Rowe in 2021). That he had upheld the flow-encouraging work-group style of Rowe was highlighted in Rosensteel's obituary: "He was kind and generous with his knowledge and his time. George brought out the best in people, challenged his students, and encouraged them to reach their potential" [104]. Hubert de Guise, Professor of Physics at Lakehead University, has stated that the qualities of Rowe in obtaining the best out of his research colleagues included that "he was unhurried by the outside world ... immensely patient" and that he had a skill "identifying, developing and leveraging the interests" of his colleagues "without impeding their progress while still maintaining sufficient focus to actually solve a non-trivial problem" [105]. Canada Research Chair in Theoretical Chemistry, Associate Professor Stijn De Baerdemacker made this comment regarding the flow experience of working with David Rowe: "David ... showed me his thought processes on multiple occasions during group meetings. I have taken that with me ... There would be a research presentation every week by one of us (or a visitor). I contributed to that on multiple occasions. The magic ingredient was that these were topics that were tangentially related to everybody's interests" [106].

Beyond investigating the role of emotional intelligence related to flow when considering career sustainability as advised regarding working with high-achieving musicians, it is suggested that HRMs study making appropriate changes to the space available to employees, their approach to time management, the open-mindedness of supervisors, and management's foundational contributions to introducing and discussing work-related problems comparable to the process developed by Rowe. The details of this process have been discussed in the 2022 publication on Rowe's methods [73]. These approaches to flow are worth studying as they have been found fruitful in achieving career sustainability in each of those physicists who participated in Rowe's research group. Furthermore, HRM would do well to investigate the publications by Csikszentmihalyi referenced in the two articles examined here that use his understanding of flow concerning their study results [18,19,82,84,88–93]. Although two of these articles by Csikszentmihalyi are specific to careers in the arts [90,92], and one is the Spanish version [82] of another publication by Csikszentmihalyi in English [14], most would be of general interest to HRM looking to develop career sustainability through work-related flow.

### 4.3. Limitations

A limitation of this work is that a decision was made by the author to search only MDPI journals regarding the keywords "flow, Csikszentmihalyi, work", rather than conducting a true scoping review (or even a systematic review) of these keywords in all journals. This represents a problem because the non-representation of comprehensive studies questions the validity of the findings The decision was made to limit the review in this way as the

aim was to investigate research published by publishing houses created with a founding commitment to sustainability because research concerning career sustainability is in its infancy as a form of human sustainability. As such, this report is intended to provide a foundation for this growing discipline based on an integral commitment to sustainability by the company publishing this type of research. For this reason, it was considered important that the integrity of publishing such research be ensured in creating this foundation. Although other publishing houses currently note their commitment to sustainability [45–47], MDPI appears to be the only publishing house that meets the stringent criterion of being created as a publishing house in this regard. Based on this exclusion criterion, a firm foundation has been provided for subsequent research contributing to the theories in this field that may include both scoping and systematic reviews.

An additional limitation of a study comparing mentions of flow, happiness, and PERMA factors in HRM-related documents is that happiness and PERMA factors were investigated only concerning their mention in studies that investigated flow. As such, there are other studies of happiness [107,108] alone and in combination with PERMA factors [109–111] that do not discuss flow. Given that the focus of this work is flow in making a comparison with happiness or PERMA factors, as only flow has been found to relate to career sustainability, this omission is to be expected. Nevertheless, it represents a limitation of this study.

There were three papers identified that investigated Csikszentmihalyi's understanding of flow and, of these, only two that did so in relation to his research findings. As a result, the work environments in which flow was considered were those of professional musicians and theoretical mathematical physicists—two exceptionally creative fields open to few employees. There is no research currently available in MDPI journals regarding flow experiences (as defined by Csikszentmihalyi) in more mundane occupations. For this reason, it is unknown if what is important regarding flow to developing sustainable careers for other employees is similar to what has been found evident within these technically difficult occupations.

The evaluation of the articles for their authors' points of view was contingent on the reading done by this author. This is an additional limitation. Although this author undertook the present study with the aim of objectivity, it is possible that the author had an unrecognized cognitive bias [112]. Various frameworks have been developed to debias research. Nevertheless, there remains little research on the efficacy of these models and, as such, how to recognize and reduce cognitive bias is identified as an area in need of additional research [113].

## 5. Conclusions

The importance of work-related flow regarding career sustainability was pioneered by Csikszentmihalyi and became popular with the publication of his 1990 book, *Flow*. Notably, the theory he initially proposed regarding flow based on his research consistently has been found valid and reliable, remaining unchanged since then in representing the optimal work-related experience. The search of the keywords "flow, Csikszentmihalyi, work" of MDPI (representing a publishing house founded regarding sustainability) publications returned 28 reports for assessment. However, although flow is a stable predictor of career sustainability, within the last five years only three of these articles have been published in MDPI journals that may be of help to HRM in developing career sustainability regarding flow. Interest in flow has remained regarding career sustainability over the years; nevertheless, research focus on career sustainability has shifted from a study of flow to investigations regarding happiness and aspects of PERMA as the predominant concerns. This has resulted in a paucity of research devoted to flow—a concept that has proven its value for career sustainability unlike either a focus on happiness which is time-dependent or PERMA factors which have been found poorly constructed. As a result, HRM is advised to redirect their development of career sustainability to work-related flow in contrast to a focus on happiness or measuring PERMA factors. From the research presented in two of these articles, how

best to do this is for HRM to (1) concentrate on recruitment of employees with the courage to be different, who possess independent thought and action, and who have high levels of self-confidence, and (2) support programs that provide employees with the appropriately designed space and time to undertake what they personally value; as well as providing supervisors who are open-minded in their approach to solving problems and able to provide foundational theoretical contributions to solving work-related problems. Studying the works by Csikszentmihalyi referenced by the researchers of these two publications would be additionally helpful in developing programs. In adopting these means, HRM may anticipate strides in developing employee career sustainability.

**Funding:** This research received no external funding.

**Institutional Review Board Statement:** Not applicable.

**Informed Consent Statement:** Not applicable.

**Data Availability Statement:** Not applicable.

**Conflicts of Interest:** The authors declare no conflict of interest.

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
