# Peer review of "Work-Related Flow in Contrast to Either Happiness or PERMA Factors for Human Resources Management Development of Career Sustainability"

_psych, doi:10.3390/psych6010021_

Round 1

Reviewer 1 Report

The article has been substantially improved and its publication is favourable.

The article has been substantially improved and its publication is favourable.

Author Response

Thank you very much to the reviewer for judging the article to be substantially improved and for stating its publication is favourable.

Reviewer 2 Report

I accept this  article for publication without any specific comments. The theory/concept of flow is acceptable in the  conditions/circumstances presented in the article.

I accept this  article for publication without any specific comments. The theory/concept of flow is acceptable in the  conditions/circumstances presented in the article.

Author Response

Thank you very much to the reviewer for accepting this article for publication without any specific comments and for finding the theory/concept of flow to be acceptable in the conditions/circumstances presented in the article.

Reviewer 3 Report

Overall, the manuscript is interesting and can stand the test of time. However, the major concern stems from the selection of articles. First, it is not clear why the author(s) confined their search to MDPI journals. While carrying out a systematic literature or meta-analysis, all the articles within the defined period should be used to get comprehensive results. The non-representation of comprehensive studies questions the validity of the findings. Second, the contributions of the study are very limited in its current form. The author(s) should elaborate on this. How are the findings contributing to the theories in this field?

No specific comments.

Author Response

Overall, the manuscript is interesting and can stand the test of time.

Thank you to the reviewer for judging that the manuscript interesting and able to stand the test of time.

However, the major concern stems from the selection of articles.

Thank you to the reviewer for clearly noting that there is a major concern and that this major concern stems from the selection of articles

First, it is not clear why the author(s) confined their search to MDPI journals.

Thank you to the reviewer for specifying that it is unclear why the author has confined the search to MDPI journals. To improve the clarity, information regarding why MDPI was the only publishing house that was considered has been moved from the Limitations section to the second last paragraph of the Introduction, where this matter is first discussed. Furthermore, phrases have been added in this paragraph, and the first line of the next, to provide additional details for this decision.

Still, in that this is a limitation to this review, it remains the first mentioned in the Limitations section. This mention has now been improved to bring greater clarity to why this limitation was found acceptable by the author.

All mentions of Systematic and Scoping with respect to reviews have now been capitalized to indicate that they are technical terms associated with PRISMA requirements, whereas, “limited” is not.

Based on these changes, the Abstract has also been similarly updated.

While carrying out a systematic literature or meta-analysis, all the articles within the defined period should be used to get comprehensive results.

Thank you to the reviewer for the concern that all articles should be searched within the defined period to get comprehensive results when carrying out a systematic literature or meta-analysis. This author agrees. As the review conducted was limited, and not a Systematic review, all articles need not be searched.

The non-representation of comprehensive studies questions the validity of the findings.

The reviewer is thanked for providing this statement. It is now part of the first limitation of 4.3. Limitations.

Second, the contributions of the study are very limited in its current form.

The reviewer is thanked for stating the contributions of the study are very limited in its current form The author accepts this criticism.

The author(s) should elaborate on this.

Thank you to the reviewer for asking that the author elaborate on this. The reason is that the author had a very limited intention of providing a stable foundation for such research based on the integrity of the publishing house searched. As only MDPI met the criterion of being founded based on a concern for sustainability, it was the only publishing house searched. With this stringent foundation as a limited search, subsequent research can now be undertaken to expand the literature on the topic of career sustainability.

How are the findings contributing to the theories in this field?

Thank you to the reviewer for questioning how the finding contribute to the theories in this field. This point of how these finding are contributing to theories in this field has now been added to the first limitation of 4.3. Limitations.

In summary, the reviewer is thanked for providing these additional concerns. In answering the reviewer, the responses have improved the clarity of the manuscript.

Round 2

Reviewer 3 Report

I appreciate the author(s) for addressing all the comments and suggestions. I have no further comments.

I appreciate the author(s) for addressing all the comments and suggestions. I have no further comments.